# Network Analysis Reveals the Molecular Bases of Statin Pleiotropy That Vary with Genetic Background

Cintya E. del Rio Hernandez,[a,b] Lani J. Campbell,[a,b] Paul H. Atkinson,[a,b] Andrew B. Munkacsi[a,b]

aSchool of Biological Sciences, Victoria University of Wellington, Wellington, New Zealand
bMaurice Wilkins Centre for Molecular Biodiscovery, Victoria University of Wellington, Wellington, New Zealand

**ABSTRACT** Many approved drugs are pleiotropic: for example, statins, whose main cholesterol-lowering activity is complemented by anticancer and prodiabetogenic mechanisms involving poorly characterized genetic interaction networks. We investigated these using the *Saccharomyces cerevisiae* genetic model, where most genetic interactions known are limited to the statin-sensitive S288C genetic background. We therefore broadened our approach by investigating gene interactions to include two statin-resistant genetic backgrounds: UWOPS87-2421 and Y55. Networks were functionally focused by selection of *HMG1* and *BTS1* mevalonate pathway genes for detection of genetic interactions. Networks, multilayered by genetic background, were analyzed for key genes using network centrality (degree, betweenness, and closeness), pathway enrichment, functional community modules, and Gene Ontology. Specifically, we found modification genes related to dysregulated endocytosis and autophagic cell death. To translate results to human cells, human orthologues were searched for other drug targets, thus identifying candidates for synergistic anticancer bioactivity.

**IMPORTANCE** Atorvastatin is a highly successful drug prescribed to lower cholesterol and prevent cardiovascular disease in millions of people. Though much of its effect comes from inhibiting a key enzyme in the cholesterol biosynthetic pathway, genes in this pathway interact with genes in other pathways, resulting in 15% of patients suffering painful muscular side effects and 50% having inadequate responses. Such multigenic complexity may be unraveled using gene networks assembled from overlapping pairs of genes that complement each other. We used the unique power of yeast genetics to construct genome-wide networks specific to atorvastatin bioactivity in three genetic backgrounds to represent the genetic variation and varying response to atorvastatin in human individuals. We then used algorithms to identify key genes and their associated FDA-approved drugs in the networks, which resulted in the distinction of drugs that may synergistically enhance the known anticancer activity of atorvastatin.

**KEYWORDS** chemical genetics, epistasis, network analysis, pleiotropy, statins, synthetic genetic array, synthetic lethality, yeast

Since their discovery more than 40 years ago, statins (1) have saved millions of lives via cholesterol reduction and prevention of cardiovascular disease by competitive inhibition of the rate-limiting 3-hydroxy-3-methyl-glutaryl-coenzyme A reductase (HMGCR) enzyme in the mevalonate pathway (2, 3) (Fig. 1). However, statins, like many drugs, are pleiotropic and affect other pathways, including those related to diabetes and tumorigenesis (4–8). Such pleiotropy may be exploited to investigate other useful properties of such drugs. Pleiotropic properties of drugs are often the consequence of complex gene network dynamics downstream of the primary target.

The yeast *Saccharomyces cerevisiae* is an established eukaryote model organism (9) in which about 70% of its genes show high conservation with humans not only in sequence but also in biological function (10). This was cleanly illustrated with humanization of yeast

Address correspondence to Andrew B. Munkacsi, andrew.munkacsi@vuw.ac.nz.

The authors declare no conflict of interest.

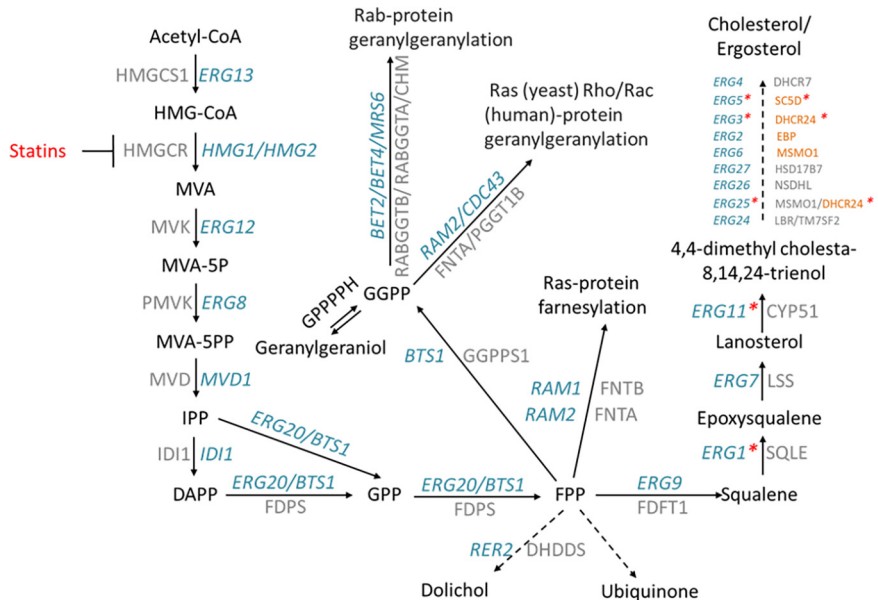

**FIG 1** Statins inhibit the synthesis of HMGCR and downstream products in the mevalonate pathway. Statins are competitive inhibitors of HMGCR encoded by *HMG1* and *HMG2* in yeast and the HMGCR gene in humans. A critical step in the mevalonate pathway is mediated by the enzyme geranylgeranyl diphosphate synthase (encoded by *BTS1* in yeast and *GGPPS1* in humans), where the main ergosterol/cholesterol-synthesis pathway branches off to synthesize other fundamental cellular components for isoprenylation of small GTPases. Genes in blue are yeast genes, and genes in gray are their human orthologues. Red asterisks in yeast genes indicate oxygen-dependent steps of the pathway. Human genes in orange at the end of the cholesterol pathway are less conserved with yeast and do not correspond to the yeast gene to the left.

(i.e., expression of human genes in yeast), where only 20% amino acid identity was required for human genes to complement the deletion of orthologous yeast genes (11). Relevant to this study, human HMGCR restored the viability of yeast lacking its two paralogue genes, *HMG1* and *HMG2* (12). Indeed, many steps of the mevalonate pathway were originally elucidated in yeast (13, 14). Because of its genetic tractability, it is a powerful aid for the study of the mevalonate pathway (13, 14), cancer cell biology (15–17) and complex phenotypes in general (18–22).

Complexity may be investigated by genetic interactions involving epistasis (23), which measures functionality shared by the interacting pairs of genes. In yeast, interactions may be scored in high-throughput screens called synthetic genetic array (SGA) analysis that measure colony size phenotype changes exerted in pairs of double gene deletion strains (24, 25) or by a gene deletion paired with a gene product inhibitory drug (26). The tractability of yeast genetics allowed genome-wide cataloguing of genetic interactions that are called synthetic lethal when a double mutant exhibits no growth or synthetic sick when the double mutant exhibits reduced growth (27, 28).

From these synthetic lethal and synthetic sick interactions, gene networks have been assembled representing 5.4 million interactions in the S288C genetic background (29, 30). Networks in turn may be analyzed for key genes using graph centrality metrics (31–34), and here we applied such methodology to statin pleiotropy. We had at hand three libraries of yeast genome-wide deletion strains constructed in three different genetic backgrounds —S288C, Y55, and UWOPS87-2421 (here referred as UWOPS87) (18, 35)—allowing us to additionally characterize statin pleiotropy by genetic background.

## RESULTS

The overall scheme of our study (Fig. 2) is to elucidate the mevalonate pathway-specific genetic interactions integral to statin bioactivity. Using SGA methodology (36), we generated 25,800 double deletion yeast strains, each lacking a gene in the statin pathway and a second gene in the yeast genome of statin-susceptible and statin-

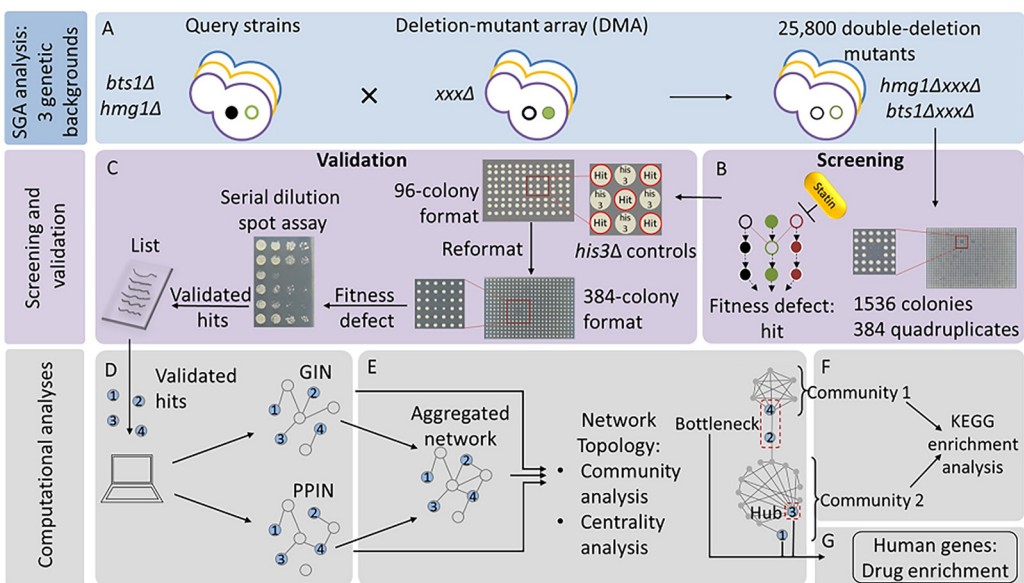

**FIG 2** Flow diagram for the methods used to identify interactions and pathways behind atorvastatin pleiotropy in three genetic backgrounds. Single deletion mutant array libraries (A) in the S288C genetic background (depicted in blue outline) and in the recently created UWOPS87 and Y55 genetic backgrounds (depicted in yellow and purple) through a backcross methodology (18) were used to generate genome-wide double deletion mutants (deletion mutant genes depicted as empty circles) as models to investigate the atorvastatin pleiotropy in three genetic backgrounds (B). About 25,800 double deletion mutants in 1,536-colony format (384 quadruplicate colonies per agar plate) were created, treated with atorvastatin, and screened to identify fitness defects that would reveal epistatic interactions as measured by decreased colony size. Atorvastatin-hypersensitive double mutants were then validated in serial dilution spot assays and used as input to create genetic (GIN) and protein-protein (PPIN) interaction networks (C). GINs and PPINs were multilayered in one network (D) per genetic background and subjected to network topology analyses. The network centrality metrics pinpointed bottleneck and hub genes of high biological relevance. The communities of genes identified through network modularity (E) were analyzed through a KEGG enrichment analysis to distinguish key metabolic pathways. Human orthologues of the key yeast genes were used in a search for drug enrichment (F) to identify potential combination therapies to enhance the anticancer activity of atorvastatin.

resistant genetic backgrounds since cholesterol-lowering activities of statins vary among individuals (15, 16). The genes within the mevalonate pathway investigated were *HMG1*, the predominantly active target of atorvastatin under aerobic conditions (about 80% of the activity compared to its paralogue *HMG2* [37]), and *BTS1*, the mediator of the off-branch pathway from the main ergosterol synthesis pathway to isoprenylation of GTPases. The double deletion mutants were treated with atorvastatin, and hypersensitive mutants were compiled into multilayer networks. Topology centrality metrics and functional enrichment in chemical genetic interaction networks were used to identify key genes and cellular processes regulating statin activity, which by definition are candidate targets to use in combination with statins to enhance their anticancer activity.

**Screening for statin-specific epistasis in genome-wide deletion libraries in three genetic backgrounds.** In order to measure the chemical genetic effects of atorvastatin and the combined *hmg1Δ xxxΔ* and *bts1Δ xxxΔ* double gene deletions, it was necessary to ensure atorvastatin was not present in excess so the statin effect and double mutant effect could be distinguished. To achieve this, we determined the concentration of atorvastatin that reduced the growth of single deletion *hmg1Δ* and *bts1Δ* mutants to 70% of normal growth in the S288C, Y55, and UWOPS87 deletion library strains. Accordingly, we separately deleted the *HMG1* and *BTS1* genes through PCR-directed mutagenesis and homologous recombination in the three backgrounds and then treated with atorvastatin to characterize the toxicity range of concentrations of the drug (Fig. 3A). All three genetic backgrounds showed the same sensitivity when *HMG1* was deleted (i.e., synthetic sick at 5 $\mu$M atorvastatin, synthetic lethal at 20 $\mu$M atorvastatin), probably because all the backgrounds are equally reliant on *HMG1* to cope with atorvastatin. Contrastingly, when *BTS1* was deleted in S288C, synthetic lethality occurred in 1 $\mu$M atorvastatin, while the same concentration exerted only a

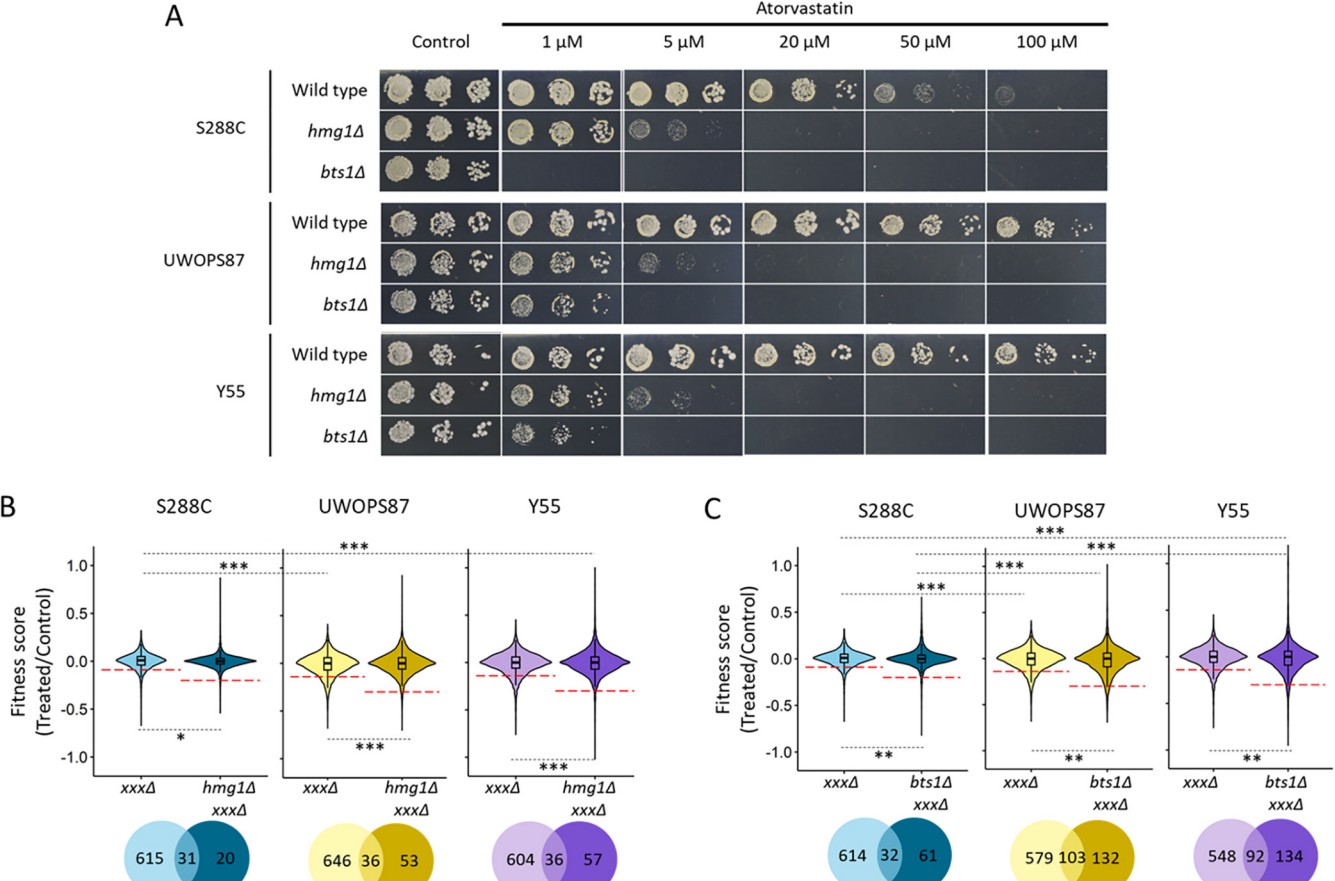

**FIG 3** Atorvastatin sensitivity confers similar synthetic sickness/lethality in strains with *HMG1* deleted and varies in strains with *BTS1* deleted in three genetic backgrounds. (A) Haploid cells deficient of *HMG1* or *BTS1* and their wild types in three genetic backgrounds were pinned on increasing concentrations of atorvastatin in serial dilution and incubated for 2 days at 30°C. (B) Violin plot distributions of average fitness of 12,900 strains as measured by colony sizes ($n = 4$) of the *xxxΔ* and *hmg1Δ xxxΔ* mutants as well as (C) *xxxΔ* and *bts1Δ xxxΔ* mutants, where positive scores represent increased fitness and negative scores represent decreased fitness. The red dashed lines indicate the score cutoff values selected for validation in independent assays for double deletions that did not overlap the *xxxΔ* single deletions. Venn diagrams visualize the overlap in the number of genes below the cutoff lines. Statistical differences were evaluated by Student's *t* test (*, $P < 0.05$; **, $P < 0.01$; ***, $P < 0.001$).

mild fitness defect in UWOPS87 and Y55. This may be because the downstream *BTS1* gene mediates several branches from the mevalonate pathway, possibly providing background-specific statin resistance pathways.

We then investigated atorvastatin-specific (triple mutant) chemical genetic hypersensitivity in our three different genetic background (S288C, Y55, and UWOPS87) double deletion libraries using the *hmg1Δ* and *bts1Δ* mutants as SGA query strains. Thus, the 70% (30% inhibitory concentration [IC$_{30}$]) concentration in all libraries, including the *hmg1Δ xxxΔ*, *bts1Δ xxxΔ*, and *xxxΔ* mutants, was chosen from the ranges 0.2 to 64 $\mu$M for the *hmg1Δ xxxΔ* mutants, 0.01 to 64 $\mu$M for the *bts1Δ xxxΔ* mutant, and 10 to 320 $\mu$M for the *xxxΔ* mutants. Using this information, *hmg1Δ xxxΔ* double deletion mutants were thus screened at 0.8 $\mu$M atorvastatin, *bts1Δ xxxΔ* mutants with double deletions in S288C were screened at 0.05 $\mu$M, and *bts1Δ xxxΔ* mutants with double deletions in Y55 and UWOPS87 were screened at 0.5 $\mu$M. The single deletion library *xxxΔ* control was screened at 9 $\mu$M for S288C, 10 $\mu$M for UWOPS87, and 35 $\mu$M for Y55. Violin plots showed that the average of scored colony sizes did not differ among the three genetic backgrounds when *HMG1* was deleted (Fig. 3B), but it did differ between S288C and the resistant genetic backgrounds when *BTS1* was deleted (Fig. 3C), thus adding evidence to our observations above that all three backgrounds are equally reliant on *HMG1*, but downstream *BTS1*-mediated pathway branches provide background-specific resistance to atorvastatin.

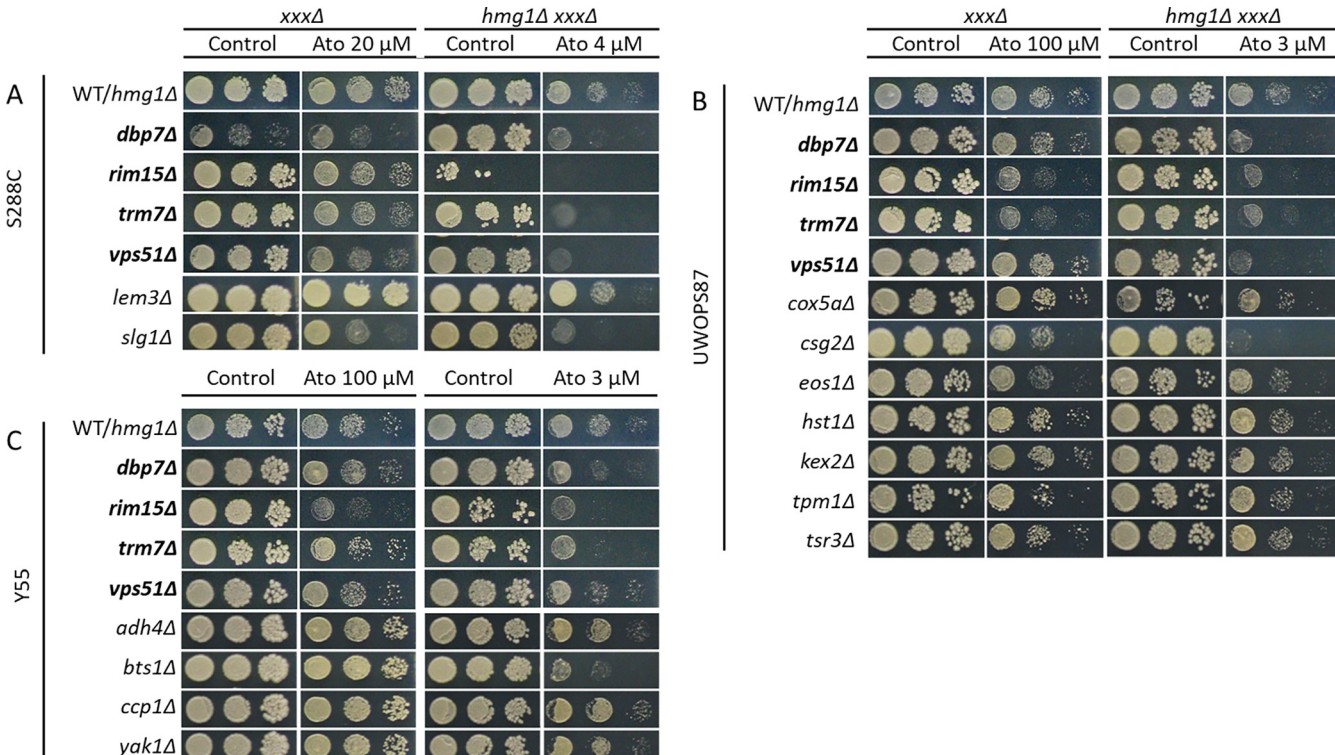

**FIG 4** Four *hmg1Δ xxxΔ* double deletion mutants were hypersensitive to atorvastatin treatment in three genetic backgrounds, while others depend on genetic background. Haploid cells derived from SGA analyses and gene deletion libraries were pinned on SC with or without supplementation of atorvastatin in serial dilution and incubated for 2 days at 30°C. Shown here are deletions of genes that enhanced sensitivity to atorvastatin treatment in the (A) S288C, (B) UWOPS87, and (C) Y55 genetic backgrounds. The WT/*hmg1Δ* panel refers to either the nonmutated wild types (WT) for the *xxxΔ* strain panels or the *hmg1Δ* single deletion mutants for the *hmg1Δ xxxΔ* double deletion strain panels. Gene deletions in boldface indicate interactions overlapping in three genetic backgrounds.

**Experimental validation of atorvastatin-hypersensitive double deletion mutants.** High-throughput screening experiments in high-density formats tend to suffer from false-positive and false-negative noisy data. To validate the atorvastatin-hypersensitive interactions, first we established a cutoff for the scored colonies (pixel-based colony size scored values assigned in SGAtools via Gitter [38]) of 3 standard deviations (SD) below the median for *hmg1Δ* strains and of 2.5 SD below the median for *bts1Δ* strains. (Thus strains with scores below −0.2 for S288C and below −0.3 for UWOPS87 and Y55 were considered genuine hypersensitive mutants.) Given our specific interest in epistatic interaction effects unique to the double deletions, strains that were sensitive in single and double deletion mutants in the presence of statins were excluded from further analysis (Fig. 3B and C). Using these cutoff criteria, we found atorvastatin-specific interactions in 20, 53, and 57 *hmg1Δ xxxΔ* strains for S288C, UWOPS87, and Y55, respectively. Likewise, for *bts1Δ xxxΔ* strains, there were 61, 132, and 134 atorvastatin-specific interactions in S288C, UWOPS87 and Y55, respectively. Atorvastatin-specific interactions were then individually validated in a second step by plating in a 384-colony quadruplicate format to confirm atorvastatin sensitivity, followed by confirmation in the *hmg1Δ xxxΔ* mutant (6 interactions in S288C, 8 interactions in Y55, and 11 interactions for UWOPS87) and *bts1Δ xxxΔ* mutant (7 interactions in S288C, 12 interactions in Y55, and 15 interactions in UWOPS87) in serial spot dilution assays (Fig. 4 and 5). Of the 40 yeast genes identified, 29 of them have human orthologues that have been previously annotated to cancer, 21 to diabetes, 10 to myopathies, and 2 to rhabdomyolysis, and 8 are known targets of statins (see Table S1 in the supplemental material).

**Construction of multilayer gene-gene and protein-protein interaction networks.** Network centrality analyses are often performed in single-layer networks—that is, connections between nodes based on one type of functional relationship. Assessment of multilayer networks has now expanded the usefulness of centrality analyses by analyzing two or more layers of interactions of different types of data (39). Similar to a single-layer

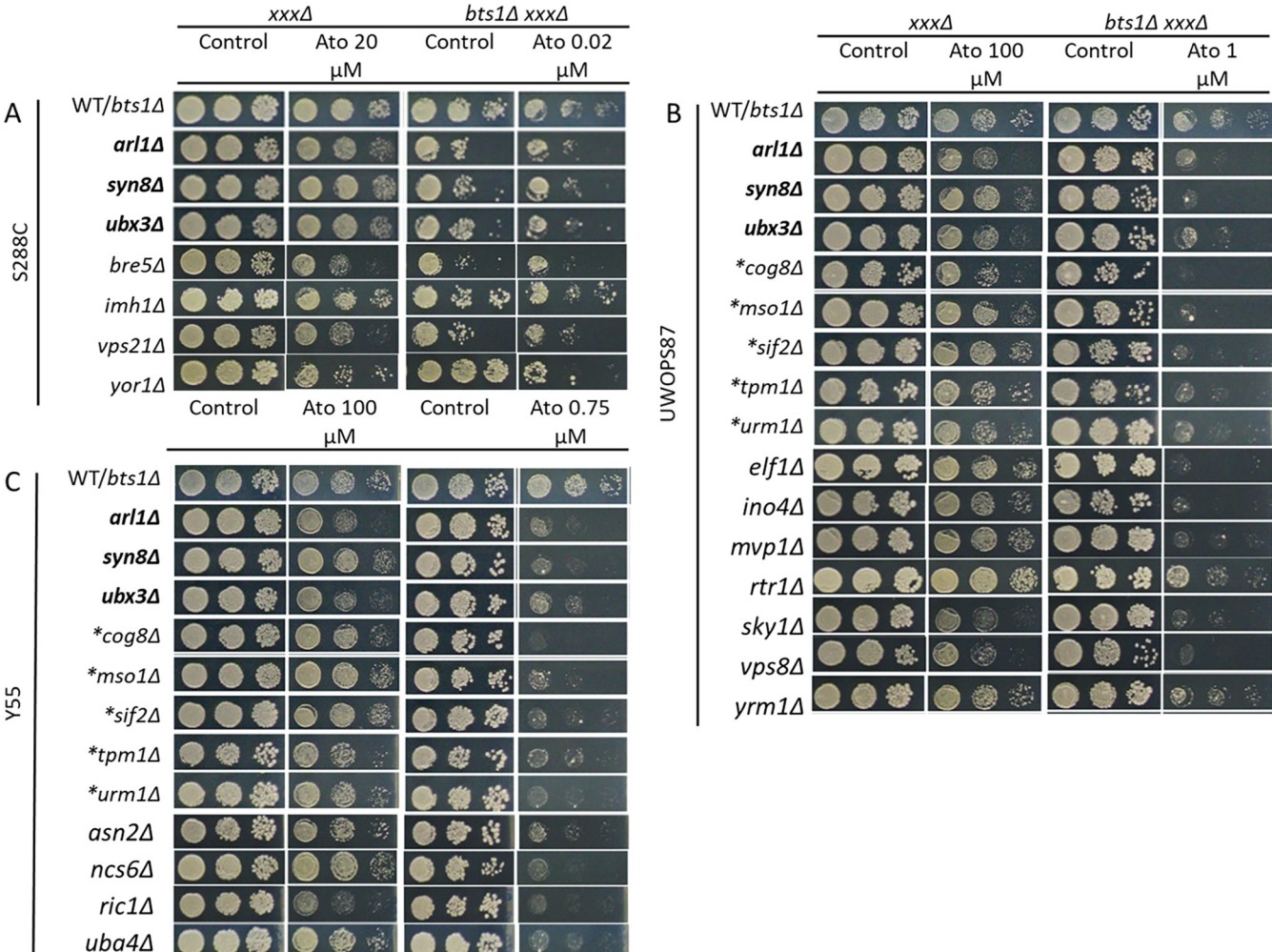

**FIG 5** Eight *bts1Δ xxxΔ* double deletion mutants were hypersensitive to atorvastatin treatment in at least two genetic backgrounds, while others depend on the genetic background. Haploid cells derived from SGA analyses and gene deletion libraries were pinned on SC with or without supplementation of atorvastatin in serial dilution and incubated for 2 days at 30°C. Shown here are deletions of genes that enhanced sensitivity to atorvastatin treatment in the (A) S288C, (B) UWOPS87, and (C) Y55 genetic backgrounds. WT/*bts1Δ* refers to either the nonmutated wild type (WT) for the *xxxΔ* strain panels or the *bts1Δ* single deletion for the *bts1Δ xxxΔ* double deletion strain panels. Gene deletions in boldface indicate interactions overlapping in three genetic backgrounds; asterisks indicate interactions overlapping in two genetic backgrounds.

network, albeit just more complex, multilayer networks are basically *n*-dimensional matrices or tensors that can be investigated using graph mathematical methodologies.

We assembled multilayer networks from the 17 and 23 genes (Fig. 4 and 5) that were validated to be interactive with *HMG1* and *BTS1*, respectively, where the list of validated genes was augmented (path length of 2) in known genetic interaction networks (GINs) (40) and protein-protein interaction networks (PPINs) (41–43). Thus, we identified and visualized GINs and PPINs specific to each type of interaction and each genetic background (Fig. 6 and 7; see Fig. S1 and S2 in the supplemental material). The numbers of nodes and edges for GINs or PPINs were generally lower than those of the multilayer network (Table 1; Table S2), demonstrating that the connectivity of multilayer networks was more robust than that of GINs and PPINs alone.

**Network topology centrality analyses identify genes critical to atorvastatin sensitivity in multilayer networks.** Networks may be analyzed for informative topological metrics called "centralities" (44), where briefly, the more central a gene is to a network, the more biological relevance it has to the phenotype. Three centrality measurements were thus calculated separately for each GIN and PPIN as well as for the combined GIN-PPIN multilayer network, namely, betweenness centrality (the shortest path length between two

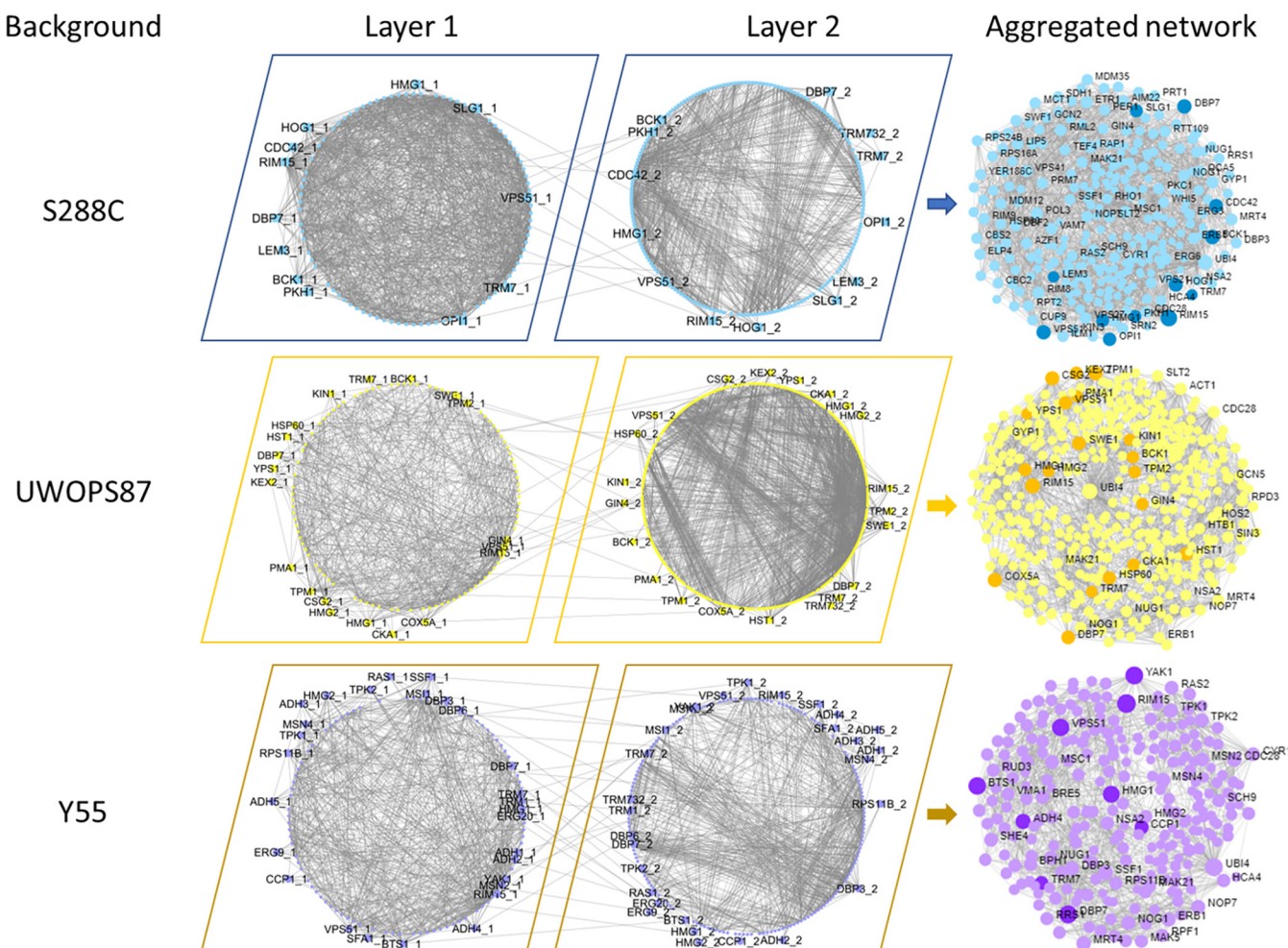

**FIG 6** Multilayer networks derived from atorvastatin-sensitive *hmg1Δ xxxΔ* interactions. GINs (layer 1), PPINs (layer 2), and the edges between them were integrated in a multilayer network using TimeNexus. Edges between layers connect overlapping nodes in the two layers, and the genes linking these edges are shown in the periphery of circular networks. Darker nodes in multilayer networks are validated atorvastatin-sensitive interactions.

nodes) (34), closeness centrality (the shortest path length between one node and all other nodes) (33), and degree centrality (number of neighbors) (32). As the GINs and PPINs exhibited different overall patterns of centrality (Fig. S3), the multilayer network was prioritized to ensure consideration of both GINs and PPINs in a common analysis (Fig. 8).

*RIM15*, a gene encoding a protein kinase involved in cell proliferation, was ranked highly in the three genetic backgrounds for betweenness, closeness, and degree centralities in the *hmg1Δ xxxΔ* networks (Fig. 8 and Table 1; Fig. S4). Impressively, this result reflects the atorvastatin hypersensitivity we found for the *hmg1Δ rim15Δ* mutant (Fig. 4). The betweenness, closeness, and degree centrality metrics for *RIM15* were 0.12, 0.49, and 93, respectively, for S288C, compared to 0.07, 0.42, and 66 for UWOPS87 and 0.09, 0.48 and 55 for Y55. Likewise, the *CDC28* kinase master regulator of mitotic and meiotic cell cycles, was also ranked highly for the three centrality metrics in the three genetic backgrounds (Fig. 8; Table 1). The involvement of kinases in statin responses points to fundamental effects of statins on aspects of metabolism other than cholesterol metabolism.

For the *bts1Δ xxxΔ* networks, the t-SNARE *TLG2* gene, which mediates the fusion of endosome-derived vesicles with the late Golgi compartment, was distinct for the three centrality metrics in Y55 and UWOPS87, while it was less distinct in S288C (Fig. 8 and Table 1; Fig. S5). The betweenness, closeness, and degree centrality metrics for *TLG2* were 0.02, 0.55 and 65, respectively, for S288C, compared to 0.02, 0.49 and 82 for UWOPS87 and 0.04, 0.50 and 84 for Y55. The ubiquitin protease cofactor *BRE5* gene,

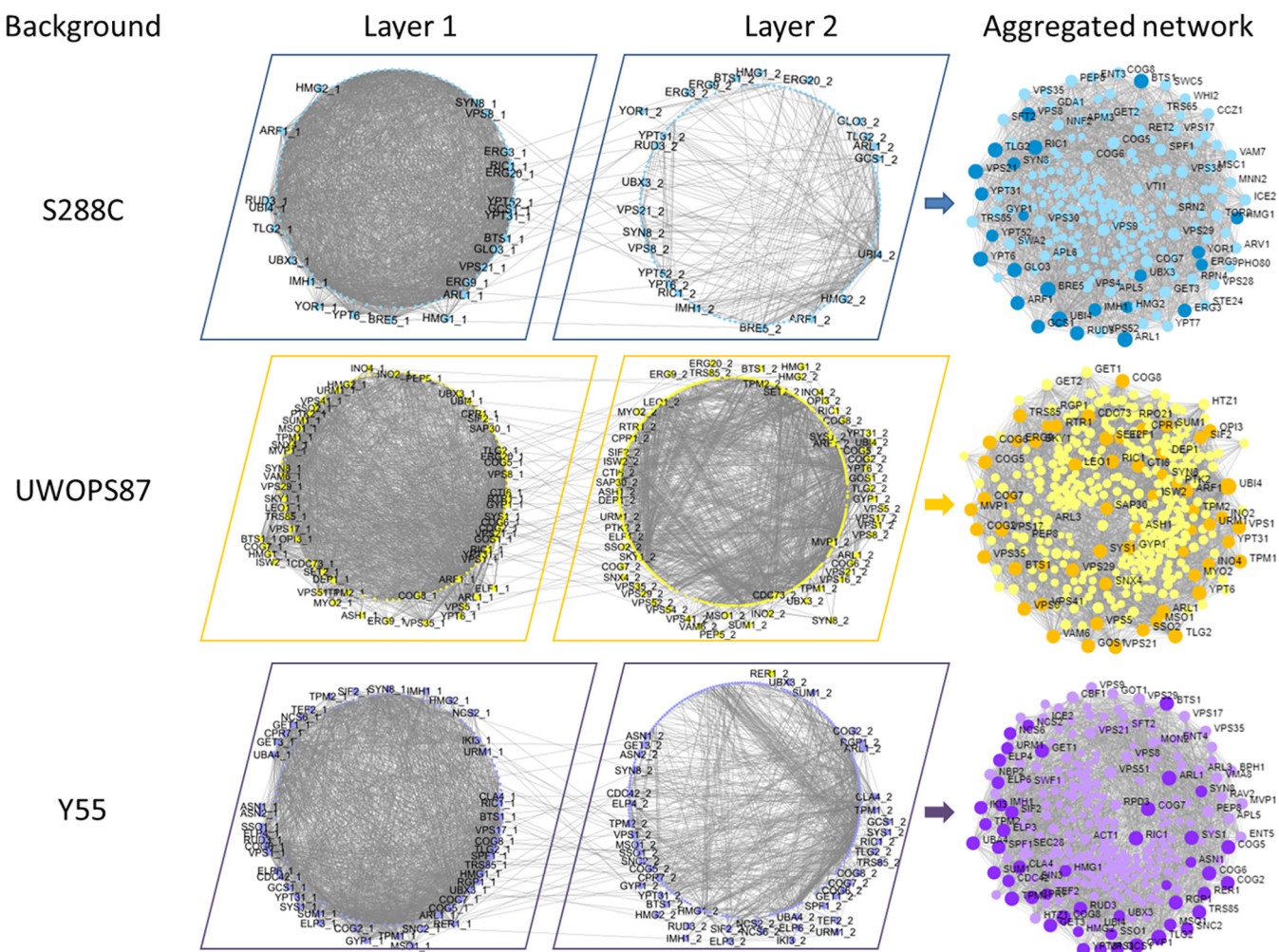

**FIG 7** Multilayer networks derived from atorvastatin-sensitive *bts1Δ xxxΔ* interactions. GINs (layer 1), PPINs (layer 2), and the edges between them were integrated in a multilayer network using TimeNexus. Edges between layers connect overlapping nodes in the two layers, and the genes linking these edges are shown in the periphery of circular networks. Darker nodes in multilayer networks are validated atorvastatin-sensitive interactions.

which coregulates, with *UBP3*, the anterograde and retrograde transport between the endoplasmic reticulum (ER) and Golgi compartments, was the highest-ranked gene for the three centrality metrics in S288C (the betweenness, closeness, and degree centrality metrics for *BRE5* were 0.13, 0.59, and 82), which is further supported by the atorvastatin hypersensitivity of the *bts1Δ bre5Δ* only in S288C (Fig. 5). In contrast, *BRE5* was not required for network connectivity in UWOPS87 and Y55 (i.e., was not present in the multilayer networks for UWOPS87 and Y55). The involvement of genes mediating vesicular fusion and transport provides insight into the diverse functions of the *BTS1* branch in the mevalonate pathway.

**Community analysis identifies functional modules in multilayer networks for three genetic backgrounds.** To gain more insight into the structural organization of the multilayer networks in order to identify metabolic pathways mediating atorvastatin sensitivity, community analysis was conducted to partition the networks to functional subnetworks (modules) that are more interconnected than random (45). We detected 3 to 6 modules in each network (Fig. 9). Each module exhibited significant enrichment for specific metabolic pathways ($P < 0.05$), and in most cases, pathways enriched in these modules did not overlap in all three genetic backgrounds (Fig. 9).

For the *hmg1Δ xxxΔ* networks, the longevity regulation pathway and its tightly linked processes autophagy and mitophagy were found enriched in all three genetic backgrounds (Fig. 9). In contrast, the longevity regulation pathway was only enriched

**TABLE 1** Top betweenness centrality measurements for multilayer networks in three genetic backgrounds

| Query gene | Genetic background | No. of nodes/edges | Gene name | Betweenness | | Closeness | | Degree | |
|---|---|---|---|---|---|---|---|---|---|
| | | | | Score | Rank | Score | Rank | Score | Rank |
| *HMG1* | S288C | 228/2,118 | *RIM15* | 0.12 | 2nd | 0.49 | 4th | 93 | 2nd |
| | | | *CDC28* | 0.03 | 7th | 0.47 | 7th | 31 | 25th |
| | | | *DBP7* | 0.11 | 3rd | 0.45 | 10th | 44 | 4th |
| | | | *HMG1* | 0.03 | 9th | 0.45 | 11th | 24 | 54th |
| | UWOPS87 | 464/2,556 | *RIM15* | 0.07 | 2nd | 0.42 | 15th | 66 | 2nd |
| | | | *CDC28* | 0.04 | 8th | 0.44 | 4th | 53 | 3rd |
| | | | *DBP7* | 0.04 | 6th | 0.40 | 44th | 36 | 14th |
| | | | *HMG1* | 0.03 | 11th | 0.43 | 9th | 27 | 32nd |
| | Y55 | 265/1,525 | *RIM15* | 0.09 | 2nd | 0.48 | 3rd | 55 | 2nd |
| | | | *CDC28* | 0.05 | 7th | 0.46 | 8th | 37 | 7th |
| | | | *DBP7* | 0.06 | 4th | 0.45 | 12th | 45 | 3rd |
| | | | *HMG1* | 0.04 | 9th | 0.46 | 6th | 33 | 12th |
| *BTS1* | S288C | 224/2,239 | *TLG2* | 0.02 | 12th | 0.55 | 3rd | 65 | 7th |
| | UWOPS87 | 428/3,743 | *TLG2* | 0.02 | 10th | 0.49 | 5th | 82 | 2nd |
| | Y55 | 300/2,767 | *TLG2* | 0.04 | 5th | 0.50 | 3rd | 84 | 2nd |

in UWOPS87 and Y55 for the *bts1Δ xxxΔ* networks (Fig. 9). Since statins extend life span, although it varies person to person, and the chronological life span in yeast mimics the postmitotic state of cancer cells (46, 47), we sought to test the importance of specific genetic interactions to statin-mediated longevity. Consequently, we measured the chronological life span of *BTS1* epistatic strains (*bts1Δ rpd3Δ*, *bts1Δ ras2Δ*, *bts1Δ rfm1Δ*, *bts1Δ sum1Δ*, *bts1Δ hst1Δ*, and *bts1Δ sir1Δ* mutants) representative of modules in UWOPS87 and Y55 that were enriched for the longevity regulation pathway. The survival integral, the area underneath the survival curve, for each single deletion strain as well as the *bts1Δ xxxΔ* strains was calculated in the presence and absence of atorvastatin over a period of 13 days (Fig. 10). For S288C, the survival area remained relatively consistent across double mutants with or without atorvastatin (Fig. 10A), suggesting treatment did not impact the survival in this genetic background. In contrast, survival was significantly increased with treatment in UWOPS87 and was more pronounced in *bts1Δ xxxΔ* double deletion mutants than single deletion mutants for the *sum1Δ*, *hst1Δ*, and *sir1Δ* strains (Fig. 10B), suggesting these specific chromatin-histone interactions with atorvastatin increase the chronological life span of UWOPS87 strains. For Y55, survival was significantly increased with treatment in the *bts1Δ rpd3Δ* double mutant compared to *bts1Δ* and *rpd3Δ* strains (Fig. 10C), revealing the importance of the histone deacetylase *RPD3* gene only in this background. These results experimentally validate the *in silico* community analyses that vary by genetic background and confirm the importance of specific chromatin-histone interactions mediating the life span extension activity of atorvastatin. As expected, metabolic pathways were enriched in the single-layer analysis but not in the multilayer analysis, and vice versa (Fig. S6 and S7), yet multilayer networks retrieved more relevant information since GINs could not be partitioned in communities possibly due to their high connectivity.

**Humanization of yeast epistasis reveals anticancer drugs for statin synergy.** Combination therapies may increase efficacy of repurposed drugs (48). To see if that is the case with statins and anticancer drugs, we identified the human orthologues of the key centrality genes identified in our yeast genomic analyses across the three genetic backgrounds (Table S3). Since most of them have been previously annotated to cancer (Table S1), we evaluated these genes for enrichment in the Drug Signature Database (49), providing greater specificity in selection of synergistic combinations (Fig. 11). This analysis detected 205 drugs with "signature genes" integral to their bioactivity as well as atorvastatin ($P < 0.05$). Of these 205 drugs, the maximum and minimum odds ratios

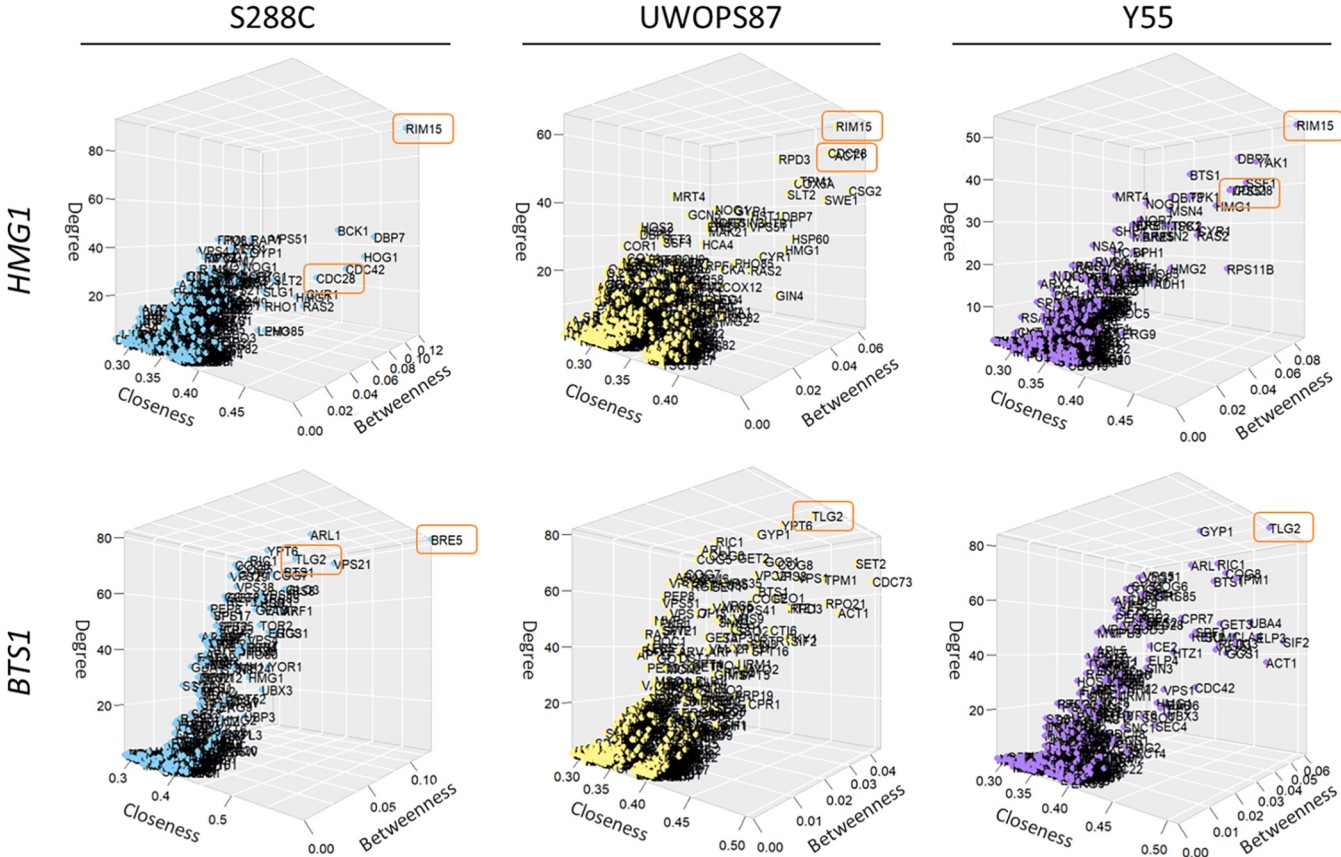

**FIG 8** Network topology centrality analyses of multilayer networks identify key *HMG1/BTS1* interactors for atorvastatin sensitivity. Centrality measurements (degree, closeness, and betweenness) were calculated for each gene and visualized in a 3D plot. *UBI4* was excluded because, due to its highly interactive nature, it skewed all the other nodes to one corner of the plot, obscuring the relevance of other genes. High-centrality genes *RIM15*, *CDC28*, *TLG2*, and *BRE5* are enclosed in orange rectangles.

were 86 and 2, respectively. To compare the chemical genetic profiles of the top-ranked drugs, the odds ratio values for the top 20 drugs by *P* values and their signature genes were visualized in a bubble plot (Fig. 11).

The 32 signature genes shown in Fig. 11 represent seven major processes targeted by specific drugs. Four drugs (docetaxel, probenecid, verlukast, and hesperetin) were correlated with ABC transporter genes involved in numerous functions, including drug efflux, and that provoke failure of chemotherapeutics (50). Fifteen drugs (GW779439X, dinaclinib, docetaxel, lestaurtinib, KW-2449, RO-31-8220, palbociclib, AZD5438, CGP74514A, sunitinib, JNK-9L, staurosporine, PKR [RNA-dependent protein kinase] inhibitor, hesperetin, and AS-59957) were correlated with kinase activity contributed by cyclin-dependent kinase (CDK) genes, dual-specificity tyrosine-regulated kinase (DYRK) genes, and mitogen-activated protein kinase (MAPK) (HPK) genes involved in cell cycle. Four drugs (lestaurtinib, palbociclib, sunitinib, staurosporine) were correlated with the MAST1 gene involved in survival signaling pathways that confers cell resistance to the chemotherapeutic cisplatin (51). Four drugs (lestaurtinib, KW2449, sunitinib, and staurosporine) were correlated with the PRPF4B gene, an essential gene for triple-negative breast cancer metastasis (52). Finally, eight drugs (lestaurtinib, KW-2449, RO-31-8220, AZD5438, GW5074, sunitinib, staurosporine, and A-674563) were correlated with serine/arginine-rich protein-specific kinase (SRPK) genes involved in activation of various signaling pathways that mediate cytotoxic effects of genotoxic agents, including cisplatin (53). The pyrazolopyridazine GW779439X ranked the highest of all drugs and compounds (*P* = 2.42E−09; odds ratio = 86), which was mainly due to key centrality genes in CDK genes identified with the *HMG1* query.

Thus, the majority of the top 20 drugs (i.e., the lowest adjusted *P* [Adj*P*] value) identified here for potential synergy with atorvastatin have exhibited anticancer activity,

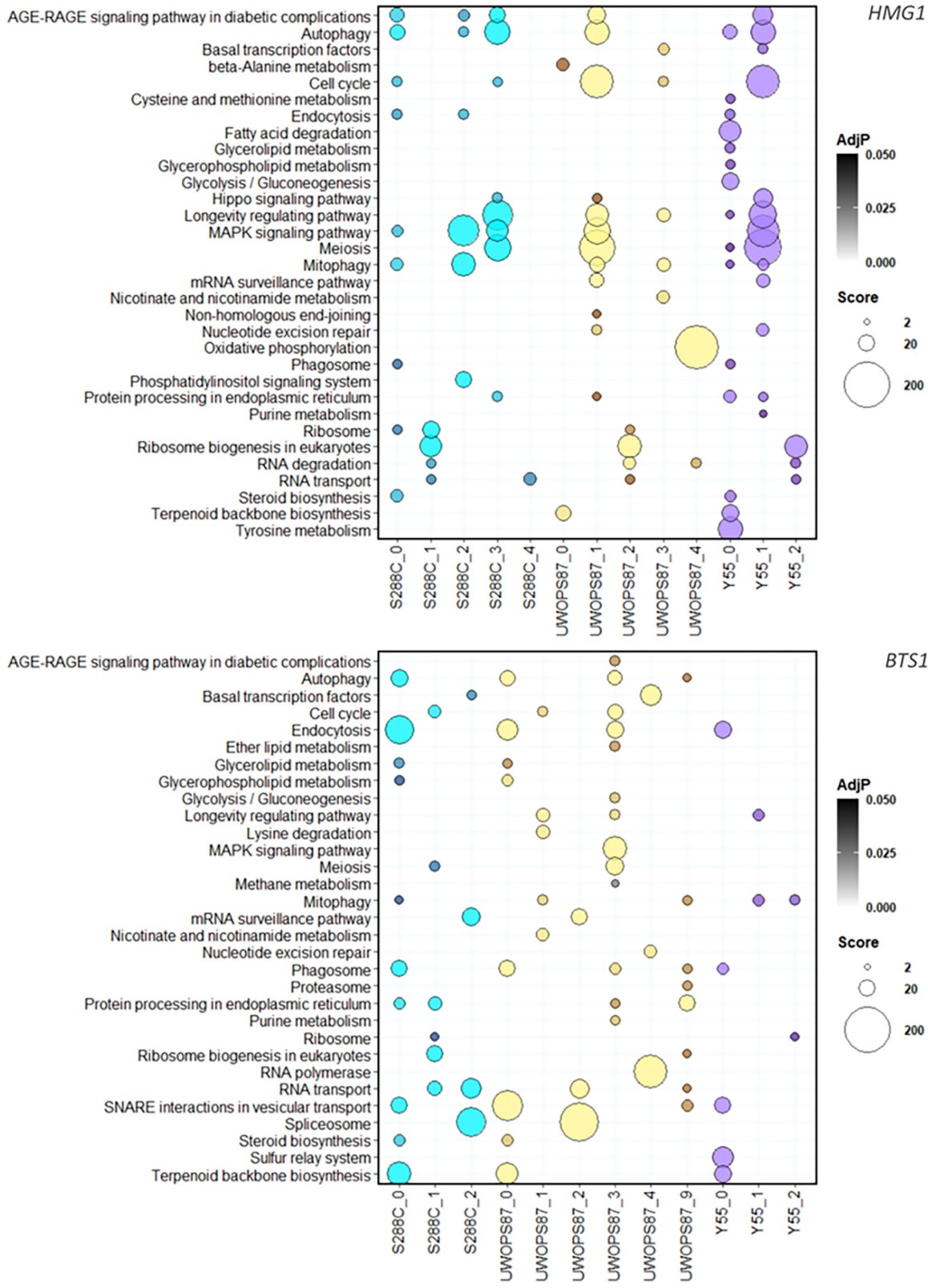

**FIG 9** Metabolic pathway enrichment of modules in multilayer networks for atorvastatin sensitivity. Bubble plots show enrichment for each of the modules (named for their genetic background) identified through community analysis for *HMG1* (top panel) and *BTS1* (bottom panel) interactions. The size of the bubbles is relative to the enrichment score for each pathway, while the intensity of the colors is relative to the adjusted *P* value. The *x*-axis labels show the genetic background followed by the number of modules. Numbers missing in the sequence are modules without significantly enriched pathways.

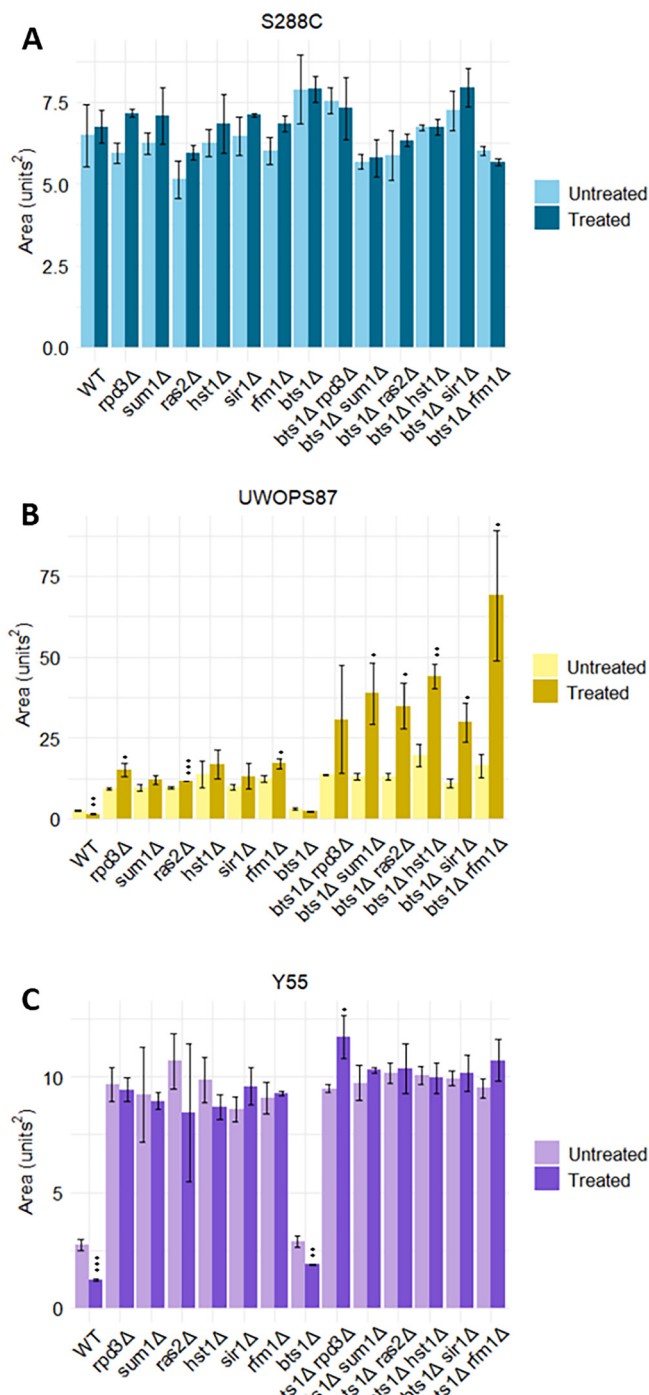

**FIG 10** Atorvastatin treatment in the UWOPS87 genetic background increases the survival integral of double mutants. Cells were grown in triplicate with and without atorvastatin. Cultures were left growing at 30°C for 2 weeks, and growth was measured every second day for a 2-week period via hourly measurements of optical density. YODA was used to calculate the surviving cell percentage. Data are shown as the mean ± standard deviation (SD) ($n = 3$). *, $P \leq 0.05$, **, $P \leq 0.01$, and ***, $P \leq 0.001$, by Student's $t$ test relative to the vehicle control.

and only 2 have been investigated for statin synergy (Table S4). These drugs with established anticancer activity include dinaciclib, docetaxel, lestaurtinib, vorinostat, palbociclib, and sunitinib. Interestingly, one of the top results is docetaxel, a well-established chemotherapeutic for the treatment of breast cancer that was previously investigated for synergy with lovastatin, albeit the trial was terminated for lack of

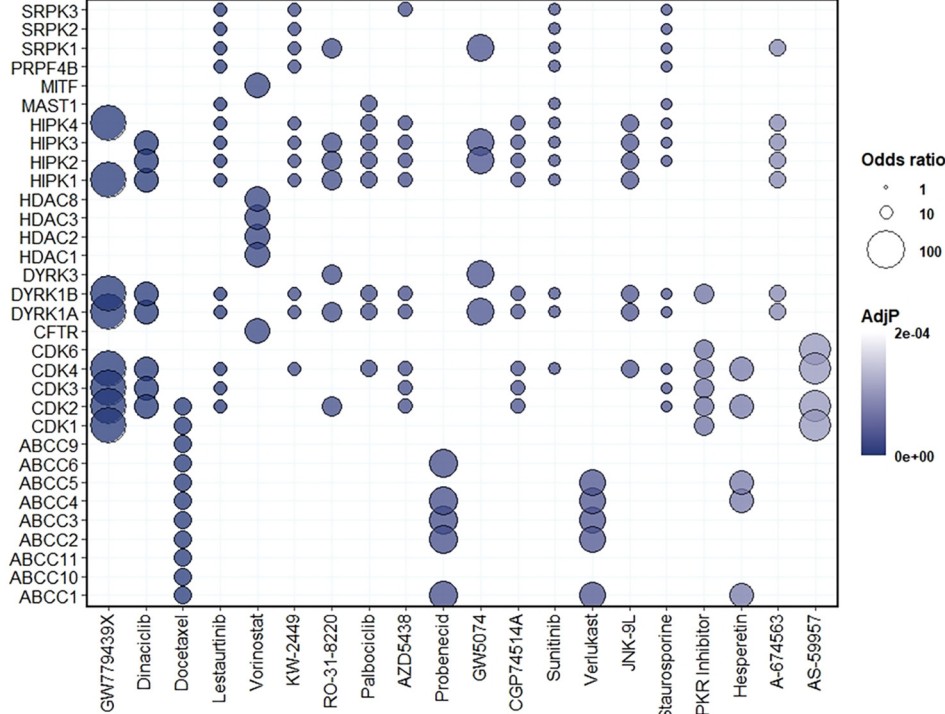

**FIG 11** Human orthologues of yeast interactions reveal drugs/compounds to test for synergy with atorvastatin. Human orthologues of validated genes and bottleneck genes were processed via an enrichment analysis for signature genes in the Drug Signature Database. Bubble plots represent the human orthologues (*y* axis) that were enriched for drugs/compounds (*x* axis). The color of each bubble is determined by the adjusted *P* value (AdjP), and the size of bubble reflects a score computed by running the Fisher exact test for random gene sets to determine the deviation from the expected rank, where bigger bubbles represent greater enrichment.

funding (NCT00584012). Another noteworthy candidate combination therapy is probenecid, which is a drug that inhibits renal excretion and would thus increase the half-life of statin drugs. Clinical trial NCT03307252 evaluated the pharmacokinetics of probenecid with a number of drugs, including rosuvastatin, but this trial did not evaluate the anticancer activity of the statin. In addition to drugs/compounds with established anticancer activity, we also propose combination therapies with GW779439X with antibiotic properties, verlukast with bronchodilator properties, and hesperetin with a wide variety of properties, including cholesterol-lowering, antioxidant, anti-inflammatory and anticancer properties.

## DISCUSSION

Drug response involves many genes whose phenotypes may be the result of epistatic genetic interactions, pleiotropy, and dependency on genetic background, which can be analyzed as multidimensional networks (Fig. 2) via topological centrality metrics and community algorithms. Here, we used yeast genome-wide deletion libraries with two genetic probes (SGA query gene *hmg1Δ* and *bts1Δ* strains) and an inhibitory drug (atorvastatin) to define colony growth phenotypes and networks in the mevalonate pathway (Fig. 1). Genetic background is known to affect genetic interactions (18, 35, 54), so we investigated genetic interactions in the standard S288C strain and two additional deletion libraries in the genetic backgrounds UWOPS87 and Y55. With the network topological centrality and community algorithms used here, clear pathways of Gene Ontology (GO) cellular processes emerged in the case of the *HMG1* or *BTS1* probes for interactions involved in autophagy, aging, endocytosis, actin and unfolded protein response (UPR) pathways. The following discusses specifics of these topics.

Statins are known to activate autophagy (55–57), yet the mechanism is not fully

understood. Here, we distinguish *RIM15* as a key statin modulator in positively regulating autophagy, since *RIM15* deficiency conferred hypersensitivity in the atorvastatin-treated *HMG1* query and *RIM15* was a high-betweenness gene (bottleneck) in three genetic backgrounds. Bottlenecks are of high relevance because they tend to connect functional clusters of genes (34, 44). We enhanced the networks derived from colony growth by including published interactors with *RIM15* of path length 2, and 75% of the genes belonged to a single community module in all of the genetic backgrounds. This community was enriched for functions involved in meiosis, longevity, and autophagy. This serine/threonine kinase *RIM15* gene is integral to statin-induced autophagy in yeast and may be conserved in mammalian cells via the human orthologue, MASTL.

Functional redundancy (58) is seen for *RIM15* in its role in actin-mediated processes and endocytosis as described here. Relatedly, we show here that *CDC28* is a top centrality gene in the *HMG1* genetic interaction networks that previously was shown to have a suppressing interaction with *RIM15* (59, 60). *CDC28* and *RIM15* cluster together in a cochaperone module for "actin and morphogenesis" (61). Although *CDC28* did not belong to a statistically significant community module in our study, 86% of the genes that interacted with *CDC28* in Y55 and UWOPS87 belonged to the community module corresponding to meiosis, cell cycle, and MAPK signaling, suggesting that networks are functionally redundant for these processes as well as actin/endocytosis. We note that human orthologues of *RIM15* code for cytoskeleton components, such as actin and the intermediate filament, which have shown to be part of the statin response (62).

*TPM1* is a bottleneck gene for UWOPS87 and Y55 treated with atorvastatin and a synthetic lethal genetic interaction with S288C in *BTS1* query. It is a major isoform of tropomyosin that binds and stabilizes actin cables (63). Statins have indeed been shown to induce cytoskeletal reorganization while increasing the levels of F-actin, important in cancer cell motility (64). Pleiotropy for statins is seen where the *CDC28* human orthologue CDK1 is downregulated by atorvastatin with anticancer activity in esophageal squamous cell carcinoma (ESCC) cells (65). Likewise, simvastatin induced $G_1$ arrest and inhibited cell growth of colorectal cancer cell lines by a mechanism that included downregulation of CDK4/cyclin D1 and CDK2/cyclin E1 (66). Simvastatin and lovastatin suppressed expression of CDK1, CDK2, CDK3, CDK4, and CDK6 in prostate cancer cells with reduced cell viability due to induced apoptosis and cell cycle arrest (67). Tropomyosin is a cancer prophylaxis drug target, which could have caused nuanced changes in general toxicity by synergistic combinations of statins and drugs aimed at tropomyosin-modifying genes, prescreened for activity in the yeast models as described here. The mechanism of action provides insight into drug synergy (68), as shown by the example here that identified probenecid for potential synergy with atorvastatin. Probenecid is prescribed for the prevention of hyperuricemia-associated gout. Given that high serum cholesterol levels have been correlated with hyperuricemia (69), it is likely that many patients worldwide are simultaneously prescribed probenecid and statins. Databases such as UK Biobank (70) might reveal whether simultaneous treatment of probenecid with statins has been associated with reduced rates of cancer.

Aging is considered one of the main risk factors for cancer development (71). Here, we show that double deletion of *BTS1* with *HST1* increased the chronological life span of the UWOPS87 strain, unlike the single deletion mutants. *HST1*, demonstrated to be synthetic sick with *HMG1* in UWOPS87, is an NAD(+)-dependent histone deacetylase gene paralogue to the human SIRT1 gene coding for sirtuin 1. Sirtuins are a family of protein deacetylases that regulate aging and longevity (72, 73). SIRT1 has indeed been described as part of the mechanism behind the antiaging effect of statins (74, 75), focusing on cell senescence. However, the role of SIRT1 in the chronological aging of nondividing cells has not been investigated. Our results point to a genetic background-dependent role, in which the presence and potential activation of yeast *HST1* by atorvastatin do not affect chronological life span. However, chronological life span is greatly increased by its double deletion with *BTS1* in the UWOPS87 background. For the Y55 background, it was the deletion of *BTS1* with *RPD3* that increased the

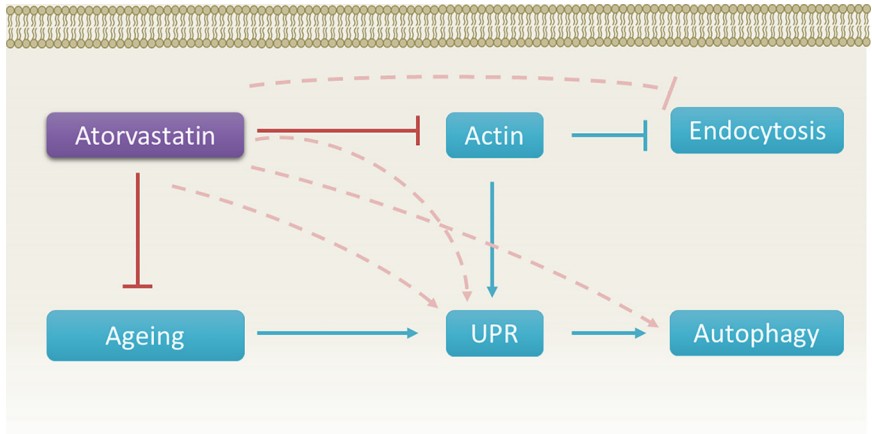

**FIG 12** Proposed integration of mechanisms. Atorvastatin inhibits components of the actin cytoskeleton, which in turn inhibits actin-mediated endocytosis and induces UPR. Atorvastatin inhibits aging pathways, which also results in the dual induction of UPR and autophagy. Hence, atorvastatin is an indirect inhibitor of endocytosis and indirect activator of UPR and autophagy. Red blunt-headed arrows point to pathways inhibited by atorvastatin. Blue arrows and the blue blunt-headed arrow point to pathways that are inhibited or induced, respectively. Dashed pink arrows and the dashed blunt-headed arrow point to inhibition or induction, respectively, of pathways via indirect mechanisms of atorvastatin.

chronological life span. *RPD3* is a histone deacetylase gene orthologue to the human HDAC1 and HDAC2 genes, which have been linked to the mechanism of anticancer activity of statins (76, 77), and HDAC1 is known to have an antiaging activity in brain cells (78). We note that statins increase the life span of the model organism *Caenorhabditis elegans* (79) and decreased mortality independent of cholesterol in humans 78 to 90 years old (47). Our results plausibly point to the importance of the *BTS1* branch with chromatin and histones in atorvastatin-mediated effects on life span.

Taken together, we have demonstrated the utility of using chemical genetics and multi-layer network analyses to elucidate genetic complexity of metabolic pathway phenotypes that may be behind drug molecular mechanisms. In this article, we discuss atorvastatin and its anticancer properties. We note that UPR/ER stress is tightly linked to autophagy (80, 81). We have previously shown UPR activation is qualitatively genetic background dependent in response to statins (18), and ER stress is also a known mechanism of the anticancer activity of statins (82). Our model here provides more information on the link between UPR and actin-mediated endocytosis (83), autophagy, and aging (84, 85). We propose that statin treatment induces the UPR, dysregulates endocytosis, and causes autophagic cell death (Fig. 12). It is plausible that all of these phenotypes have a role in the anticancer activity of atorvastatin via induction of UPR, especially since statins inhibit and remodel actin cytoskeleton (86, 87), actin is necessary for endocytosis (88), and UPR is induced in yeast mutants deficient in actin-mediated steps in endocytosis (83). The many genetic interactions involving these cellular processes described here potentially provide lists for drug targets.

## MATERIALS AND METHODS

**Yeast strains, plasmids and media.** The *S. cerevisiae* strains used in this study are described in Table S5 in the supplemental material. Stocks were stored at −80℃ in 15% glycerol. Strains that contained the URA3_CEN plasmid were grown on agar with 1 mg/mL of 5-fluoroorotic acid (5-FOA) (Kaixuan Chemical Co.) to select for uracil auxotrophs before construction of the query strains. Gene deletion libraries were maintained in synthetic complete (SC), synthetic dropout (SD), enriched sporulation, or yeast-peptone-dextrose agar as previously described (89). The media and solutions used included agar, amino acids, peptone, yeast extract, yeast nitrogen base (Formedium), ampicillin, atorvastatin calcium, glucose, monosodium glutamate, potassium acetate (Sigma-Aldrich), Geneticin sulfate, L-canavanine sulfate, *S*-aminoethyl-L-cysteine hydrochloride (thyalisine) (Carbosynth), nourseothricin sulfate (Werner BioAgents), and hygromycin B (Life Technologies). All antibiotics and supplement stocks were filter sterilized with 22-$\mu$m-pore filters (Jet Biofil).

**Synthetic genetic array analysis.** Synthetic genetic array (SGA) analysis was conducted in quadruplicate as previously described (18, 36), using a 1,536-colony format in three genetic backgrounds (S288C, UWOPS87, and Y55) with newly constructed query deletion strains, in which *HMG1* and *BTS1*

were replaced with the NATMX antibiotic resistance gene via PCR-mediated disruption, using specific primers and cycle conditions (Table S6). The plasmids used for this study were conserved in *Escherichia coli* (DH5$\alpha$) and stored at $-80°C$, including the MX4-natR switcher cassette p4339 (36). PCR products were then transformed into a *MAT$\alpha$* SGA starter strain via homologous transformation as previously described (90), and integration into the genome was confirmed by PCR as previously described (24). Plates were replica plated with an automated RoToR HDA system (Singer Instruments).

**Genome-wide growth analysis.** The selected double deletion mutant libraries (*hmg1$\Delta$ xxx$\Delta$* and *bts1$\Delta$ xxx$\Delta$*) were pinned on SC agar, incubated at 30°C overnight, and used as an inoculum source to pin on SC agar with and without $IC_{30}$ concentrations of atorvastatin that were determined for each genetic background. These plates were incubated at 30°C for 12 and 24 h, time points when the colonies were imaged using a digital camera (Canon). The colony sizes were quantified and scored through SGAtools (91), where Z-scores were used to compare growth with and without atorvastatin. (Zero indicates no difference between the control and treatment, negative scores indicate reduced fitness with atorvastatin, and positive scores indicate increased fitness with atorvastatin.) All SGA scores were visualized in violin plots generated in R, and based on their point of inflection, the cutoffs were selected to identify strains for experimental validation in 384-colony format and serial dilution spot assay.

**Validation of negative genetic interactions in the 384-colony format.** The validation of negative genetic interactions was performed in a two-step process. First, 96-colony-format plates were arrayed containing no more than 29 atorvastatin-hypersensitive double mutants each with *his3$\Delta$* control border strains and also *his3$\Delta$* control strains surrounding each candidate to ensure the colony sizes were not biased. Each plate also included a wild-type strain. The atorvastatin-hypersensitive double mutants for S288C, Y55, and UWOPS87, the *hmg1$\Delta$ xxx$\Delta$* or *bts1$\Delta$ xxx$\Delta$* strains, which did not overlap the single deletion *xxx$\Delta$* mutant, were arrayed as described. Control single deletions were also arrayed to confirm that negative interactions pertained to double deletions only. The arrayed plates were screened with the same $IC_{30}$ concentrations of atorvastatin used in the 1,536-colony format. Plates were incubated at 30°C for 24 h and imaged using a digital camera, and growth was quantified using SGAtools (91) as described above for the 1,536-colony format, and hypersensitive strains were then selected for an additional experimental validation step through serial dilution spot assays.

**Validation of negative genetic interactions in serial dilution spot assay.** Overnight cultures were prepared in 96-well plates, and four 1:10 serial dilutions were spotted using a manual pinning tool on SC agar with and without an $IC_{30}$ concentration of atorvastatin. Plates were incubated at 30°C for 48 h, imaged using a digital camera, and evaluated visually for atorvastatin-specific growth defects. A cutoff for growth defect was determined as one spot less of atorvastatin-treated versus nontreated strains and of the double deletion compared to the single deletions (query gene deletion and *xxx$\Delta$*). Those atorvastatin-hypersensitive double mutants that were validated in spot assays were then submitted to another round of spot assays, this time including the three genetic backgrounds.

**Single-layer network analyses.** Validated genetic interactions that enhanced the hypersensitivity to atorvastatin were examined in the context of gene-gene and protein-protein interaction networks. The list of validated genes was augmented with gene-gene interactions using GeneMania (40) with all available studies with a maximum number 110 interacting genes. Using NetworkAnalyst (42, 43), the list of validated genes was augmented with protein-protein interactions using the STRING database (41), which includes text mining, genomic information, coexpression, and orthology, with the additional requirement for experimental evidence with a confidence score cutoff of 900. The resulting protein-protein interaction network was a first-order network representing the input nodes with their direct interactors (path length 1), which was then augmented into a second-order network to include nodes that connected the input genes as well as nodes that were interactors (of path length 2), but which only included the minimum number of nodes necessary to maintain connectivity of the network (minimum network). The gene-gene interaction networks (GINs) and the protein-protein interaction networks (PPINs) were then integrated into a single multilayer network using TimeNexus (92) in Cytoscape (93).

**Topology centrality analysis.** The single-layer and multilayer networks were analyzed for various measurements of network centrality using the NetworkAnalyzer for undirected networks application in Cytoscape (31). Three centrality measurements were calculated: (i) degree centrality, which computes the number of edges linked to each node so that a node with degree 5 has 5 edges associated, that is, it is linked to 5 other nodes (32); (ii) closeness centrality, which corresponds to the average shortest path length of one node to every other node computed by the Newman method (33), where 0 means an isolated node and 1 is the highest centrality and connectivity; and (iii) betweenness centrality, which is the probability of passing through a node when using the shortest path length between two nodes and is computed with the highly precise algorithm developed by Brandes (34) to distinguish nodes critical to maintain a network. The three measurements of centrality were visualized as three-dimensional (3D) plots using R (94).

**Community analysis.** Functional modules (communities) in the single-layer and multilayer networks were determined using the InfoMap algorithm (95) in NetworkAnalyst (42). Statistical significance for each module was evaluated for their clustering significance or network connectivity as computed by a Wilcoxon rank sum test ($P < 0.05$).

**Pathway enrichment analysis.** Modules were investigated for their function via metabolic pathway enrichment analysis using the Kyoto Encyclopedia of Genes and Genomes (KEGG) pathway database (96) implemented in Enrichr (97, 98). Pathway enrichment was statistically evaluated using an adjusted $P$ value with the Benjamini-Hochberg method for correction (99), a Z-score reflecting the deviation of a Fisher exact test from an expected rank, and a combined score that is the product of the natural

logarithm of the $P$ value multiplied by the Z-score. Fold enrichment and the $P$ value ($<0.05$) for statistically significant pathways in each module were visualized in bubble plots using R (100).

**Chronological life span assay.** The effect of atorvastatin on the chronological life span of *S. cerevisiae* was assessed as previously described (101), with alterations. A single colony from single and double deletion strains across all three genetic backgrounds was inoculated into 5 mL of SC medium and incubated overnight at 30°C with constant agitation. Fifty microliters of each culture was removed and added to fresh tubes with and without atorvastatin, and then the mixture was incubated at 30°C with constant agitation over the course of the experiment. Outgrowth assays were conducted at various time points of 1, 3, 5, 7, 9, and 11 days where 10 $\mu$L from each culture was removed and added to a Biofill 96-well plate with 140 $\mu$L SC medium. Optical density (OD) measurements of the plate were taken using an Envision 2102 Multilabel plate reader (Perkin Elmer) at 590 nm hourly for 48 h, while the experimental tubes were placed back in the rotator at 30°C. Data were visualized using OD as a measure of viability over time and analyzed using the Yeast Outgrowth Data Analysis program (YODA) as previously described (102) to calculate doubling time inflection, time shifts, and the survival integral for each mutant and treatment over the experimental period.

**Gene set enrichment for drug signatures.** Human orthologues of genes that interact with *HMG1/BTS1* query strains as well as highly ranked centrality genes were determined using Yeastmine in the Saccharomyces Genome Database (103) and examined for significant enrichment ($P < 0.05$) in the Drug Signature Database (49) implemented in Enrichr (97, 98, 104).

**Data availability.** The data sets produced in this study are available in Tables S7 to S9 in the supplemental material.

## SUPPLEMENTAL MATERIAL

Supplemental material is available online only.
**SUPPLEMENTAL FILE 1**, PDF file, 9.1 MB.

## ACKNOWLEDGMENTS

Funding from a Cancer Society of New Zealand Wellington Division CT Collins Ph.D. Scholarship (to C.E.d.R.H.) is greatly appreciated, and this research was supported in part by the Maurice Wilkins Centre for Molecular Biodiscovery Flexible Research Program (to P.H.A.), for which we are also very grateful.

We thank Bede Busby for the Y55 and UWOPS87 deletion libraries.

A.B.M., C.E.d.R.H., and P.H.A. designed the study. C.E.d.R.H. completed all experiments and analyses except for the chronological life span assays completed by L.J.C. A.B.M., C.E.d.R.H., and P.H.A. wrote the manuscript. All authors read and approved the final version.

We declare no conflict of interest.

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
