## [Reviewer comments · Microbiology Spectrum]

Microbiology Spectrum

Genome-wide network analysis in *Saccharomyces cerevisiae* reveals the molecular bases of statin pleiotropy that vary with genetic background

Cintya Del Rio Hernandez, Lani Campbell, Paul Atkinson, and Andrew Munkacsi

Corresponding Author(s): Andrew Munkacsi, Victoria University of Wellington

Review Timeline:

Submission Date:	October 13, 2022
Editorial Decision:	December 22, 2022
Revision Received:	February 15, 2023
Accepted:	February 18, 2023

Editor: Robert Arkowitz

Reviewer(s): Disclosure of reviewer identity is with reference to reviewer comments included in decision letter(s). The following individuals involved in review of your submission have agreed to reveal their identity: Walaa K. Mousa (Reviewer #2)

Transaction Report:

DOI: <https://doi.org/10.1128/spectrum.04148-22>

December 22, 2022

Dr. Andrew Brett Munkacsi
Victoria University of Wellington
School of Biological Sciences
Wellington
New Zealand

Re: Spectrum04148-22 (Genome-wide network analysis in *Saccharomyces cerevisiae* reveals the molecular bases of statin pleiotropy that vary with genetic background)

Dear Dr. Andrew Brett Munkacsi:

I apologize for lengthy review process. As you will see both reviewers found the work to be well written, the experiments to be comprehensive and carefully designed. Reviewer #1 raised some minor issues (in the attached pdf) that should be able to be easily addressed. To expedite matters it should be possible for me to make a quick decision on the revised manuscript without sending it back to the reviewers.

Link Not Available

Sincerely,

Robert Arkowitz

Journals Department
Reviewer comments:

Reviewer #1 (Comments for the Author):

Cintya and his/her colleagues provide a comprehensive analysis study of the atorvastatin pleiotropy effect through high-throughput SGA screening together with downstream gene network analysis. It is interesting in application of the *S. cerevisiae* genetic analysis model to investigate the potential properties of the atorvastatin. The experimental design is reasonable using both statin-sensitive and resistant *S. cerevisiae* strains for a fast screen. The main concern I have in the article reading is the

concentrations selection of atorvastatin in the screening of those isolates among three S.c genetic backgrounds. Since this is the foundation of the whole paper for downstream gene screening and analysis, I suggest illustrate the process and result in detail.

Reviewer #2 (Comments for the Author):

The research presented in this manuscript is very interesting with carefully designed experiments and smooth writing style that appeals not only to specialist but to a broader readers. I have no major comments.

Staff Comments:

Preparing Revision Guidelines

Please return the manuscript within 60 days; if you cannot complete the modification within this time period, please contact me. If you do not wish to modify the manuscript and prefer to submit it to another journal, please notify me of your decision immediately so that the manuscript may be formally withdrawn from consideration by Microbiology Spectrum.

Cintya and his/her colleagues provide a comprehensive analysis study of the atorvastatin pleiotropy effect through high-throughput SGA screening together with downstream gene network analysis. It is interesting in application of the *S. cerevisiae* genetic analysis model to investigate the potential properties of the atorvastatin. Since atorvastatin has antifungal activity, the experimental design is reasonable using both statin-sensitive and resistant *S. cerevisiae* strains for a fast screen. The main concern I have in the article reading is the concentrations selection of atorvastatin in the screening of those isolates among three *S.c* genetic backgrounds (e.g. described in lines 137-143). Since this is the foundation of the whole paper for downstream gene screening and analysis, I suggest illustrate the process and result in detail.

General comments:

1. Line 127 and Fig 3A, the author concluded the *hmg1Δ* synthetic sick at 5 μ M atorvastatin, synthetic lethal at 50 μ M atorvastatin. However, the showed Conc. of atorvastatin used are 1 μ M, 5 μ M, 50 μ M, and 100 μ M. There is a big atorvastatin concentration gap between 5 to 50 μ M, it is necessary to narrow the concentration gaps here, e.g. 5 μ M, 15 μ M, 45 μ M, 90 μ M; or 5 μ M, 25 μ M, 50 μ M.
2. Line 143-148: from the fig 3B, I can see a significant difference between S288C and the resistant strain Y55 or UWOPS87 for *hmg1Δ* from the violin plot, please explain.
3. Line 158 and fig 3B, the figure contains both violin plot and venn plot for *hmg1Δ xxxΔ* and *bts1Δ xxxΔ* strains. To have a better visualization and understanding for readers, I suggest plot the *hmg1Δ xxxΔ* and *bts1Δ xxxΔ* strain results in left and right panel, respectively, and the Venn plot was put bottom in each panel.
4. Line 162-167, rearrange the sentences to make it clear how the conclusion (i.e., 6 interactions in S288C, 8 interactions in Y55 and 11 interactions for UWOP287 in *hmg1Δ xxxΔ*) is achieved. Same as the conclusion numbers in *bts1Δ xxxΔ*.
5. Fig 4: The *hmg1Δrim15Δ* double mutant seems have growth defect especially in S288C background strain. I am wondering whether it is true rim15 affect the drug sensitivity of *hmg1Δ*.
6. Line 180: change Figs 6 and 7 to Figs 4 and 5.
7. Lines 192-196: separate the sentence into two.
8. Line 209, highlight *RIM15* and *CDC28* in fig 8 as colours. Same as *TLG2* and *BRE5*.

9. Line 225, which is further supported by the atorvastatin hypersensitivity of the *bts1Δ bre5Δ* only in S288C (Fig 5). It is essential to Mark the number of degree centrality metrics in three isolates for comparison and consistency of the hypersensitivity measurement result
10. Line 242, remove the space before the bracket.
11. Line 247, remove (Fig 9).
12. Line 30: Statin treatment induces the unfolded protein response. I did not see any results related to unfolded protein response (UPR) pathways in the whole article. Indeed the relative result is from the author's previous publication (Busby et al, 2019), rearrange the sentence to clarify.

VICTORIA UNIVERSITY OF
WELLINGTON
TE HERENGA WAKA
NEW ZEALAND

School of Biological Sciences

Te Wāhanga Pūtaiao

VICTORIA UNIVERSITY OF WELLINGTON, PO Box 600, Wellington 6140, New Zealand

Phone +64 4 463 5171 Email andrew.munkacsi@vuw.ac.nz

26 January 2023

Microbiology Spectrum
c/o ASM Journals
1752 N St. NW
Washington, DC 20036
Attn: Dr. Christina Cuomo, Editor-in-Chief

Dear Dr. Cuomo:

On behalf of our coauthors, we are resubmitting our paper “Network analysis reveals the molecular bases of statin pleiotropy that vary with genetic background” as an Article in *Microbiology Spectrum*.

We greatly appreciate the constructive comments by the reviewers. We have taken suggestions by the two reviewers very seriously and have responded to all concerns via the completion of adjustments to the writing and figures of the manuscript. This has very much enhanced the manuscript providing further scientific validation of the pleiotropic effects of the commonly prescribed cholesterol-lowering atorvastatin.

As requested, we assented to all editorial suggestions and incorporated these (highlighted in yellow) in our resubmitted manuscript. Our detailed responses to the individual comments are below.

Thank you and the reviewers for feedback that has undoubtedly improved the manuscript. We look forward to your instructions as to how we can submit high-resolution images of our figures.

Sincerely,

Andrew Munkacsi, Ph.D.
Assistant Professor of Chemical Genetics
Victoria University of Wellington
School of Biological Sciences
Alan MacDiarmid Building, Room 322
Wellington 6012
New Zealand
Email: andrew.munkacsi@vuw.ac.nz

Reviewer #1 remarks to authors:

1. Line 127 and Fig 3A, the author concluded the *hmg1A* synthetic sick at 5 μ M atorvastatin, synthetic lethal at 50 μ M atorvastatin. However, the showed Conc. Of atorvastatin used are 1 μ M, 5 μ M, 50 μ M, and 100 μ M. There is a big atorvastatin concentration gap between 5 to 50 μ m, it is necessary to narrow the concentration gaps here, e.g. 5 μ M, 15 μ M, 45 μ M, 90 μ M; or 5 μ M, 25 μ M, 50 μ M.

Our response: We have modified Figure 3A as requested with the addition of images we had available for 20 μ M atorvastatin. This figure now includes 1, 5, 20, 50 and 100 μ M that is consistent with the requested concentrations of 5, 25 and 50 μ M.

2. Line 143-148: from the fig 3B, I can see a significant difference between S288C and the resistant strain Y55 or UWOPS87 for *hmg1A* from the violin plot, please explain.

Our response: While visual inspection of the plots show that the spread of the data pertaining to S288C differs from that of UWOPS87 and Y55, the statistical test used herein, t-test, compares the means between data rather than the spread. We deemed this test appropriate given that it is a standard method to determine the overall effect of a treatment on a given population, in this case, gene deletion mutants. The two-sample test was done assuming unequal variances. The word 'distribution' (line 141) was changed to 'average', which we hope will clarify this point.

3. Line 158 and fig 3B, the figure contains both violin plot and venn plot for *hmg1A xxxA* and *bts1A xxxA* strains. To have a better visualization and understanding for readers, I suggest plot the *hmg1A xxxA* and *bts1A xxxA* strain results in left and right panel, respectively, and the Venn plot was put bottom in each panel.

*Our response: With the addition of one concentration of atorvastatin as explained in comment 1, we have rearranged the figure to address the suggestion to have *hmg1A xxxA* and *bts1A xxxA* strain results as left and right panels, respectively (now Figures 3B and 3C) with the Venn plot at the bottom of each panel.*

4. Line 162-167, rearrange the sentences to make it clear how the conclusion (i.e., 6 interactions in S288C, 8 interactions in Y55 and 11 interactions for UWOP287 in *hmg1A xxxA*) is achieved. Same as the conclusion numbers in *bts1A xxxA*.

Our response: We realise how the arrangement of the sentence was confusing. The sentence now reads:

*"Atorvastatin-specific interactions were then individually validated in a second step by plating in 384-colony quadruplicate format to confirm atorvastatin-sensitivity, followed by confirmation in *hmg1A xxxA* (6 interactions in S288C, 8 interactions in Y55 and 11 interactions for UWOPS87) and *bts1A xxxA* (7 interactions in S288C, 12 interactions in Y55 and 15 interactions in UWOPS87) in serial spot dilution assays (Figs 4 and 5)."*

5. Fig 4: The *hmg1A rim15A* double mutant seems have growth defect especially in S288C background strain. I am wondering whether it is true rim15 affect the drug sensitivity of *hmg1A*.

*Our response: The authors acknowledge the Reviewer's concern. While the double deletion *hmg1A rim15A* exerts a growth defect on its own, with a more marked effect in S288C, we believe the atorvastatin-induced lethality has to be more than simply an additive effect, especially given that the lethality observed in the treated *hmg1A rim15A* came from a 4 μ M treatment compared to 100 μ M for *rim15A* and 20 μ M for *hmg1A*.*

*While it is possible that this sensitivity is not unique to atorvastatin, the broad function of the RIM15 kinase may well result in *hmg1A rim15A* being sensitive to other compounds. Unfortunately, we do not have any data to address this possibility.*

6. Line 180: change Figs 6 and 7 to Figs 4 and 5.

Our response: We appreciate the careful spotting of this error and have made this correction.

7. Lines 192-196: separate the sentence into two.
Our response: There are already two sentences that comprise lines 192-196, and as these sentences are clear, no change was made.
8. Line 209, highlight *RIM15* and *CDC28* in fig 8 as colours. Same as *TLG2* and *BRE5*.
Our response: We appreciate the suggestion. The figure and figure legend has been amended to highlight the genes.
9. Line 225, which is further supported by the atorvastatin hypersensitivity of the *bts1Δ bre5Δ* only in S288C (Fig 5). It is essential to Mark the number of degree centrality metrics in three isolates for comparison and consistency of the hypersensitivity measurement result.
Our response: We have included the centrality metrics for S288C in the main text (Line 220). We have also added a sentence (Lines 223-225) clarifying that BRE5 was not present in the UWOPS87 and Y55 networks.
10. Line 242, remove the space before the bracket.
Our response: We have made this correction.
11. Line 247, remove (Fig 9).
Our response: We have made this correction.
12. Line 30: Statin treatment induces the unfolded protein response. I did not see any results related to unfolded protein response (UPR) pathways in the whole article. Indeed the relative result is from the author's previous publication (Busby et al, 2019), rearrange the sentence to clarify.
Our response: We have resolved the confusion that UPR results were a part of our study. We deleted the mention of UPR from the abstract.

February 18, 2023

Dr. Andrew Brett Munkacsi
Victoria University of Wellington
School of Biological Sciences
Wellington
New Zealand

Re: Spectrum04148-22R1 (Genome-wide network analysis in *Saccharomyces cerevisiae* reveals the molecular bases of statin pleiotropy that vary with genetic background)

Dear Dr. Andrew Brett Munkacsi:

Thank you for addressing all the concerns raised by the reviewers.

Your manuscript has been accepted, and I am forwarding it to the ASM Journals Department for publication. You will be notified when your proofs are ready to be viewed.

Sincerely,

Robert Arkowitz
Editor, Microbiology Spectrum
